# A Practical Data Extraction, Cleaning, and Integration Method for Structural Condition Assessment of Highway Bridges

**Gongfeng Xin** [1], **Fidel Lozano Galant** [2], **Wenwu Zhang** [1], **Ye Xia** [3,*] and **Guoquan Zhang** [3,*]

1   Shandong Hi-Speed Group Innovation Research Institute, Jinan 250014, China
2   Department of Civil Engineering, Universidad de Castilla-La Mancha, Av. Camilo Jose Cela s/n,
    13071 Ciudad Real, Spain
3   Department of Bridge Engineering, Tongji University, Shanghai 200070, China
*   Correspondence: yxia@tongji.edu.cn (Y.X.); zgqtj@tongji.edu.cn (G.Z.)

**Abstract:** The success of regional bridge condition assessment, a crucial component of systematic maintenance strategies, relies heavily on comprehensive, well-structured regional bridge databases. This study proposes the data extraction, cleaning, and integration method for the construction of such databases. First, this research proposes an extraction method tailored for unstructured data often present in inspection reports. Additionally, this paper meticulously outlines a cleaning procedure designed to rectify two distinct categories of typical errors that are present within the inspection data. Subsequently, this study takes a holistic approach by establishing integration rules that harmonize data from various sources, including inspection records, monitoring data, traffic statistics, as well as design and construction blueprints. The architectural framework of the regional bridge information database is then meticulously laid out. To validate and demonstrate the effectiveness of the method, this study applies them to a set of representative highway bridges situated within Shandong Province. The results show that this approach can be used to successfully establish a functional regional bridge database. The database plays a pivotal role in harnessing the latent potential of an extensive range of multi-source information and propels the field of bridge condition assessment forward by providing a solid basis for informed decision making and strategic planning in the realm of infrastructure maintenance.

**Keywords:** regional bridge information database; regional bridge condition assessment; data mining; bridge inspection reports

## 1. Introduction

Regular bridge inspections are vital for assessing bridge structures. They provide detailed records of bridge conditions. Critical bridge monitoring offers additional regional insights [1]. However, the current system uses a 'one bridge, one file' approach. This means data from each bridge are used only for its own assessment and repair. This method overlooks possible similar patterns of deterioration in different bridges. It also increases the complexity and cost of managing bridges. Bridges in a regional network are interconnected, not isolated [2,3]. Over the years, a vast amount of data has been collected on these bridges. It is important to integrate and standardize these data into a road network bridge database. Such integration would enhance the maintenance of the bridge network. It could also help predict future structural conditions within the network [4,5].

In recent years, global scholars have focused on regional bridge management and utilization. Precise evaluation is key to effective bridge management. To this end, governments worldwide have developed various codes for bridge assessment. Italy, for example, has guidelines for risk classification and bridge safety assessment [6]. Similarly, other regulations include the "Specifications for Maintenance of Highway Bridges and Culverts" [7] and China's "Standards for Technical Condition Evaluation of Highway Bridges" [8]. Beyond traditional methods, novel machine-learning algorithms are now being used. These

algorithms detect defects and compute health indices, offering faster and more accurate evaluations [9–13]. A well-structured Bridge Management System (BMS) is essential for storing and using these assessment outcomes effectively.

Bridges are managed at two levels: bridge level and network level. In this study, they are referred to as "single bridge level" and "network level." Reliance on a single management model is inefficient, especially for small and medium span bridges. These models often treat bridges as isolated entities and use disconnected inspection methods [14]. Regional bridges share attributes, making network-level evaluation and management essential.

Globally, Bridge Management Systems (BMS) are the main methods for managing multiple bridges. The National Bridge Inventory (NBI) database [15] created by the Federal Highway Administration (FHWA) in 1968, was the precursor to modern BMS. Originally developed for data management, the NBI has evolved to include evaluation, prediction, and decision making due to increased maintenance needs and limited funding [16]. The UK, Japan, and Denmark have similar systems like NATS [17], J-BMS [18], and DANBRO [19]. China's development in this area has been slower, mainly using the China Highway and Bridge Management System (CBMS) and Shanghai's bridge management system.

The Bridge Management System (BMS) has greatly improved bridge management. Yet, it has imperfections and practical applications reveal deficiencies. A key BMS function is recognizing bridge deterioration patterns. This transforms raw data into domain-specific knowledge, crucial for network-level evaluation [20]. As described in reference [21], this involves linking structural condition to independent variables.

There are two main approaches to this link: historical health data and probabilistic reliability models (PRM) [22]. The first predicts future conditions using past deterioration data. The second forecasts deterioration by analyzing time-varying factors like corrosion. PRMs have been successful in both stationary and non-stationary process analysis [23–26]. However, PRM is better suited to univariate cases. Multivariate scenarios pose challenges due to complexity and computational demands. In contrast, historical health data approaches handle multivariate situations well [27,28]. They use extensive historical data to track deterioration over a bridge's life. The accuracy of these models depends on the amount of available data. Health monitoring systems are costly; therefore, inspection reports are often the primary data source. However, the quality of these reports can vary, making data cleaning and integration crucial for future analyses.

To this date, few studies have comprehensively addressed the form, acquisition methods, integration rules, and information data structure regarding the multi-source information pertaining to regional bridges. The demands of network-level evaluation underscore the need for significant improvements in many aspects.

This paper details integration rules and a data structure for a regional bridge information database, designed to maximize the use of extensive multi-source data. It is divided into three main sections: Section 2 describes the framework for information integration, covering data sources, key features, expression formats, and rules. Section 3 discusses methods for extracting and cleaning data from inspection reports, the primary data source for regional bridges. Finally, Section 4 provides a practical example of this integration, using data from 812 highway bridges in Shandong Province.

## 2. Information Integration for Regional Bridges

### 2.1. Inherent Characteristics of Information Integration

The sources and integration of bridge group information for network-level assessments exhibit inherent characteristics:

1.  Hierarchical decomposition: Both bridges and road networks are intricate systems composed of various substructures. The database should implement a multi-tiered decomposition and traceability of the road network, bridges, and components, ensuring the feasibility of retrieval between different levels, thus providing the data logic foundation for network-level assessments.

2.  Interactions: The deterioration of individual bridge components mutually influences one another. For instance, cracks in the bridge deck can expedite the deterioration of primary beams, while damaged bearings can lead to deck impairments. Consequently, the database should encompass the interactions among these components to unveil the inherent connections between structural performance and condition changes.
3.  Universality and scalability: The database should be designed to meet the current analytical requirements while ensuring its capacity for future expansion and updates, accommodating the inclusion of new data or integration of novel analysis methodologies.
4.  Representation of time-varying data: Bridge data are commonly categorized into static data, such as span length and materials, and dynamic time-varying data, such as bridge age and traffic volume. Dynamic data are often associated with changes in structural characteristics. Therefore, the database should support dynamic representation to elucidate the spatiotemporal relationships within the data.

*2.2. Data Source*

The inspection data serve as the paramount repository of bridge-related information within a region. Thus, the inspection report stands as the most immediate and relatively comprehensive historical record of these data, typically archived in the format of a technical condition assessment sheet. A typical assessment sheet comprises three fundamental components:

1.  The fundamental parameters of bridges encompass key fields, including the affiliated road section, bridge pier location, bridge length, primary span configuration, span length, date of construction, and inspection date, among others.
2.  Component-level assessment data, including superstructure, substructure, and deck system, along with corresponding ratings.
3.  Overall bridge rating and maintenance records, including the overall assessment of the bridge's condition and its maintenance records.

The described data collectively characterize crucial information about the bridge, such as its structural features, service life, spatial distribution, environmental influences, and condition assessments. These insights are of paramount significance in understanding the deterioration process of the inspected structure.

Simultaneously, the transportation network serves the functions of passage and cargo transportation. From the perspective of structural loading during the service life of bridges, traffic loads constitute the most prominent external influence. Given that the inspection reports do not record traffic flow information for the specific road segment where the inspected bridge is situated, it becomes imperative to incorporate additional data sources. To address this, the collection of annual average daily traffic volume data from various traffic monitoring stations along the highway corridor is recommended. These data can be selected based on the equivalent passenger car unit, which effectively characterizes the average traffic flow impact on the inspected bridge. Alternatively, dynamic weighing systems or video monitoring can be employed to obtain information regarding the traffic flow of the network. Notably, within vehicle classification statistics, the quantity and proportion of heavy-duty vehicles have a more pronounced impact on the structure and thus, warrant special attention.

Furthermore, the design and construction blueprints of the bridge can supplement and validate the fundamental information within the database, including structural design parameters, lane quantities, and other pertinent details.

*2.3. Multi-Source Data Logic Expression*

In the case of the aforementioned integrated multi-source data, it is essential to store and represent this information in an "attribute" format. An attribute comprises an attribute name, referring to a specific characteristic, and an attribute value, which represents the data associated with that characteristic. Different attributes may have various data formats, such

as numerical, ordinal, or nominal. Among these, numerical attributes offer the most intuitive and quantifiable means for comparison. Conversely, other formats involve qualitative expressions. Ordinal attributes, although meaningful for comparison, may pose challenges in precisely quantifying differences between successive values. Nominal attributes, linked to specific names or categories, are solely used for categorical representation. In the context of a road network database, typical attributes and their format definitions are as follows:

1. Region name, route code, bridge code, component code, and bridge pier number are all nominal attributes, signifying the spatial distribution of bridge entities. The divisions of the region should align with the specific road network characteristics, taking into consideration geographical locations and temperature–humidity patterns.
2. Year of construction, inspection date, and bridge age are all numerical attributes, representing temporal information about the bridge. Bridge age can be derived by subtracting the year of construction from the inspection date.
3. Bridge type, bridge length, maximum span, cross-sectional dimensions, design code, and design office: Bridge type, design code, and design office are nominal attributes, while the others are numerical attributes, indicating the structural characteristics of the bridge entity.
4. Annual Average Daily Traffic (ADT) and Annual Average Daily Truck Traffic (ADTT) are both numerical attributes, denoting the traffic flow the bridge accommodates. ADT is calculated as the total equivalent passenger car units, while ADTT is the sum of heavy-duty trucks and large passenger vehicles. Because ADTT primarily reflects the heavy loads that significantly contribute to the deterioration of bridges, it is chosen to represent the load information of bridges in the multi-source information database.
5. Highway classification, design load, roadbed width, and number of lanes: Highway classification and design load are ordinal attributes, while the rest are numerical attributes, illustrating the bridge's traffic capacity.
6. Overall score, overall rating, superstructure rating, substructure rating, deck rating, and ratings for various component technical conditions, as well as the deflection history of the bridge: The scores and the deflection history are numerical attributes, while the ratings are ordinal, signifying the structural state of the bridge.

Among these attributes, the majority are static in nature. However, inspection date, bridge age, ADT, ADTT, structural ratings, and scores are dynamic data. These dynamic data elements are organized as key-value pairs, associated with a temporal dimension. Each key represents a corresponding time coordinate, and the value corresponds to the attribute's value at that coordinate. It is essential to store them in accordance with their respective formats. For instance, for a bridge with a "year of construction = 2015", the data could include "bridge age = {2016:1, 2017:2}" and "overall score = {2016:99, 2017:98}".

*2.4. Data Integration*

The specific organization and data structure for information at different hierarchical levels within the region exhibit distinct characteristics. Drawing upon the principles of object-oriented programming, we introduce the concepts of "classes" and "instances."

A class abstracts specific entities, expressing a particular concept. Within the regional bridge network, there are three levels of classes: component class, bridge class, and route class. These levels exhibit a hierarchical relationship, with, for example, the route class encompassing bridge classes and bridge classes containing numerous component classes. Each class can be distinguished by its name, such as the "primary beam" class and "bearing" class, each storing distinct information. The attributes defined earlier are systematically integrated into these classes based on their applicable objects and scope, creating templates for various components, bridges, and routes.

Correspondingly, instances serve as the carriers of entities within the database. They are created based on the templates specified by the classes, representing real-world objects, such as component instances, bridge instances, and route instances, and are populated with attribute data according to their characteristics and properties. For example, Bridge A and

Bridge B are both created based on the "beam bridge" category, but they are independent bridge instances, with the former having a bridge length attribute value of 15 m and the latter having a length of 30 m. Figure 1 summarizes the integration rules for the aforementioned dataset.

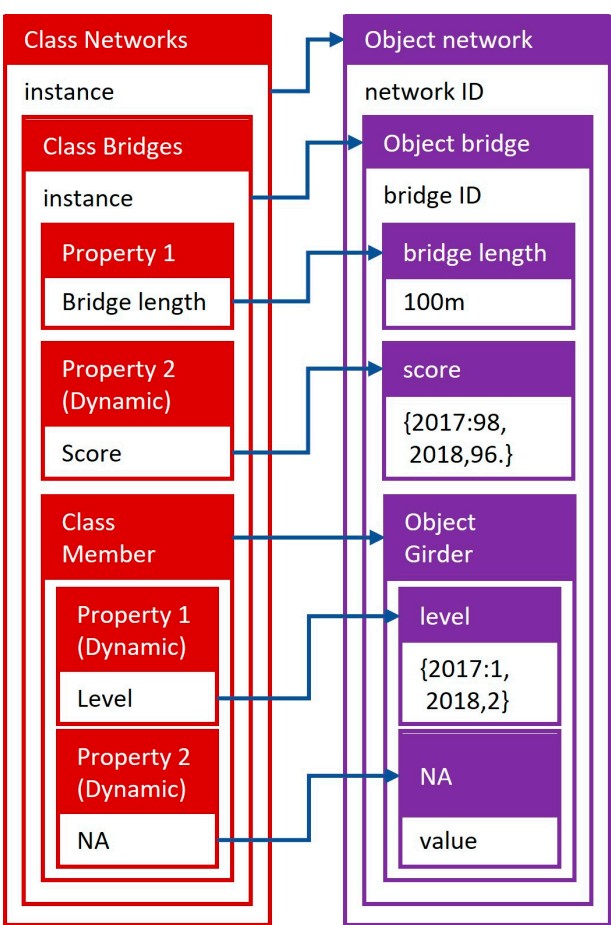

**Figure 1.** Data Integration Rules and Structures.

However, the integrated results cannot be directly employed for subsequent analysis and processing due to disparities in the original data quality from various sources. These disparities manifest as follows: (1) Physical records, with limited preservation periods, are susceptible to issues like loss and damage, affecting data continuity and accuracy. (2) Electronic records, influenced by varying management levels and execution practices, result in challenges in unifying formats and standards. (3) Errors during data entry are inevitable, leading to issues like missing or inconsistent information. Therefore, it is imperative to clean the integrated data to eliminate noise and facilitate subsequent pattern recognition.

## 3. Inspection Data Extraction and Cleaning Methods

### 3.1. Storage Format of Existing Inspection Reports

Over the past two decades, bridge inspection in China has been governed by two key standards: "Specifications for Maintenance of Highway Bridges and Culverts" (JTG H11-2004) [7] and "Standards for Technical Condition Evaluation of Highway Bridges" (JTGT H21-2011) [8]. Prior to the introduction of JTGT H21-2011 in 2011, JTG H11-2004 guided bridge inspection practices, with JTGT H21-2011 assuming this role after 2011.

JTG H11-2004 and JTGT H21-2011 employ differing approaches to bridge assessment. Both standards mandate defect inspections of bridge components. However, JTGT H21-2011 goes a step further by assigning structural condition scores to the inspected components, a

requirement not present in JTG H11-2004. Disparities also emerge in calculating the overall bridge structural condition score. In the context of JTG H11-2004, this score is determined using Equation (1):

$$D_r = 100 - \sum_{i=1}^{n} R_i W_i / 5 \tag{1}$$

In Equation (1), $D_r$ represents the overall bridge structural condition score, $R_i$ signifies the extent of defect in a component, and $W_i$ stands for the weight of the respective component. In contrast, within JTGT H21-2011, the overall bridge structural condition score is computed as the weighted average of component structural condition scores, as demonstrated in Equation (2):

$$D_r = \sum_{i=1}^{n} D_i W_i \tag{2}$$

In Equation (2), $D_r$ denotes the overall bridge structural condition score, $D_i$ represents the structural condition score of a component, and $W_i$ signifies the weight of the corresponding component.

These differing approaches to bridge assessment engender variations in the storage format of inspection reports. Reports adhering to JTG H11-2004 and JTGT H21-2011 conventions contain inspection data stored within structural condition evaluation forms. However, as depicted in Figure 2, the structure of these forms diverges significantly. In the figure, the slash "/" represents a place where information does not need to be filled in. This happens when a bridge does not have the listed component in the table.

| Structure | Index | Component | Weight | Technical condition of component | | Technical condition of structure | |
|---|---|---|---|---|---|---|---|
| | | | | score | level | score | level |
| Super Structure | 1 | Primary component | 0.70 | 100.0 | 1 | 98.1 | 1 |
| | 2 | Other component | 0.18 | 89.4 | 2 | | |
| | 3 | Bearing | 0.12 | 100.0 | 1 | | |
| Substructure | 4 | Wing wall and ear wall | 0.02 | 100.0 | 1 | 97.2 | 1 |
| | 5 | Slope | 0.01 | 100.0 | 1 | | |
| | 6 | Pier | 0.33 | 100.0 | 1 | | |
| | 7 | Abutment | 0.33 | 91.4 | 1 | | |
| | 8 | Footing | 0.31 | 100.0 | 1 | | |
| | 9 | Bed | / | / | / | | |
| | 10 | Regulating Structure | / | / | / | | |
| Deck | 11 | Deck pavement | 0.44 | 100.0 | 1 | 99.0 | 1 |
| | 12 | Expansion joint | 0.28 | 100.0 | 1 | | |
| | 13 | Sidewalk | / | / | / | | |
| | 14 | Guardrail | 0.11 | 91.0 | 2 | | |
| | 15 | Drainage | 0.11 | 100.0 | 1 | | |
| | 16 | Lighting and Sign | 0.06 | 100.0 | 1 | | |
| Overall Condition Score | Dr=97.9 | Overall Condition Level | | 1 | | | |

| Pile number | K369+295 | | Bridge name | K369+295 Channel Bridge R | |
|---|---|---|---|---|---|
| Component | weight | Degree of defect | Impact on service | Correction | Final degree of defect |
| Wing wall, Ear wall | 1 | 0 | 0 | 0 | 0 |
| Slope | 1 | 0 | 0 | 0 | 0 |
| Abutment and footing | 23 | 1 | 1 | 1 | 3 |
| Pier and footing | 24 | 0 | 0 | 0 | 0 |
| Footing scouring | 8 | 0 | 0 | 0 | 0 |
| Bearing | 3 | 0 | 0 | 0 | 0 |
| Primary bearing component | 20 | 0 | 0 | 0 | 0 |
| Other bearing component | 5 | 0 | 0 | 0 | 0 |
| Deck pavement | 1 | 0 | 0 | 0 | 0 |
| Connection | 3 | 0 | 0 | 0 | 0 |
| Expansion joint | 3 | 0 | 0 | 0 | 0 |
| Sidewalk | 1 | 0 | 0 | 0 | 0 |
| Guardrail | 1 | 1 | 1 | 0 | 2 |
| Illumination and sign | 1 | 0 | 0 | 0 | 0 |
| drainage | 1 | 0 | 0 | 0 | 0 |
| Regulating structure | 3 | 0 | 0 | 0 | 0 |
| Other | 1 | 0 | 0 | 0 | 0 |
| Overall rating | Dr=85.8 | Technical condition level | | two | |

**Figure 2.** Sample Structural Condition Evaluation Forms from Bridge Inspection Reports.

The divergence observed in inspection reports guided by distinct standards necessitates tailored extraction methods. Furthermore, it underscores the imperative of conducting thorough data cleaning processes.

### 3.2. Data Extraction and Storage

Given that inspection information resides within structural condition evaluation forms within inspection reports, the fundamental approach to data extraction involves systematically traversing all forms within the inspection report document and extracting information from forms designated as structural condition evaluation forms.

The criteria for identifying these forms can vary based on the formatting of the documents. In this study, the number of columns and rows served as criteria for judgment, with content in specific table cells also considered, such as cells containing labels like "Disease name," which often signifies the presence of disease-related information.

Multiple strategies were proposed for storing the extracted data. Employing a 2D Table Structure proves beneficial for efficient data storage. While an SQL database offers more advanced data processing and querying capabilities, for the sake of accessibility and ease of use, a 2D table structure was adopted for data storage.

The inspection data, originally structured as structural condition evaluation forms in inspection reports, assumes a table format. During data storage, a structural condition evaluation form for a specific bridge is compressed to create a corresponding row of data. Consequently, within the ultimate data storage table, each row signifies the inspection data for an individual bridge, while columns represent distinct inspection attributes for that bridge. Data pertaining to different years are segregated into separate tables.

The design of the final 2D table structure for data storage was crafted with ease of processing in mind, facilitating seamless reading and manipulation by Python's DataFrame module. Nonetheless, even users without programming expertise can interact with the data storage table using common office software like Excel 2016.

### 3.3. Data Cleaning

There are mainly two types of errors: record errors and mismatched data.

### 3.3.1. Record Errors Cleaning

Record errors typically stem from clerical mistakes made by maintenance staff when composing inspection reports. One common type of record error involves typographical inconsistencies, such as the structural condition score "94.6" being mistakenly entered as "94..6," which includes a duplication of decimal points and other similar errors.

To rectify such errors, we leverage the regular expression string matching method, which enables the identification and correction of these inaccuracies, ensuring the integrity of the actual information.

Another prevalent type of record error is known as the "Same thing, different names" error. This error arises when diverse expressions are utilized to denote identical elements. For instance, while previous inspection reports might refer to cracks as "Crack," a particular inspection report might term some of these cracks as "Rift." To address this issue, we establish dictionaries containing various expressions used to describe identical items across inspection reports. During the extraction process, we harmonize these expressions by standardizing them using the names stored within the dictionaries. This approach mitigates confusion and guarantees consistency across the data.

### 3.3.2. Mismatched Data Cleaning

The term "mismatched data" refers to instances where inspection data adhering to two distinct standards, namely JTG H11-2004 and JTGT H21-2011, do not align. In general, inspection reports following JTG H11-2004 lack component structural condition scores, which are indispensable for constructing the deterioration model of regional bridges. To address this discrepancy and ensure data uniformity, we establish conversion rules to derive structural condition scores for JTG H11-2004-guided inspection reports using available data.

When dealing with data under JTG H11-2004, the recorded information pertains to the degree of defect in different components, rather than structural condition scores. Thus, the initial task involves generating structural condition scores from the degree of defect under the JTG H11-2004 scoring system. Referring to Equation (1), the degree of defect corresponds to points deducted from the bridge's structural condition score. As the sum of all component weights in Equation (1) equals 100 (as depicted in Equation (3)), the

component weights must take the value of 100 to ensure consistency when calculating the component's structural condition score (as illustrated in Equation (4)).

$$\sum_{i=1}^{n} W_i = 100 \tag{3}$$

$$D_i = 100 - \frac{100}{5} \times R_i \tag{4}$$

In Equation (4), $D_i$ denotes the component's structural condition score, while $R_i$ stands for the degree of defect in the corresponding component. Consequently, one unit of the degree of defect translates to a deduction of 20 points from the component's structural condition score.

Subsequently, converting the structural condition score from the JTG H11-2004 and JTGT H21-2011 scoring system becomes necessary. Since the criteria for categorizing structural conditions remain uniform across both systems, establishing conversion rules based on equivalent structural condition levels is reasonable. These levels are discrete functions of the structural condition score, as outlined in Table 1.

**Table 1.** Comparison of Structural Condition Score Ranges Under Different Scoring Systems.

| Structural Condition Level | Structural Condition Score for JTGT H21-2011 | Structural Condition Score for JTG H11-2004 |
|:---:|:---:|:---:|
| Level 1 | [95, 100] | [88, 100] |
| Level 2 | [80, 95] | [60, 88] |
| Level 3 | [60, 80] | [40, 60] |
| Level 4 | [40, 60] | [0, 40] |
| Level 5 | [0, 40] | \ |

Let $l(x)$ symbolize this function. However, the discreteness of $l(x)$ introduces inaccuracy in describing the actual structural condition. For example, components with scores of 94.9 and 95.1 under the JTGT H21-2011 scoring system would fall into structural condition levels of class 2 and class 1, respectively, even though their actual structural conditions exhibit minimal differences. In light of this, a new function $L(x)$, was introduced, creating a continuous structural condition level. $L(x)$ was devised as a polyline function, interpolating linearly between the left points of each interval in Table 1. The construction of $L(x)$ under both scoring systems is depicted in Figure 3. In the figure, the red points stand for the interval endpoint as shown in Table 1.

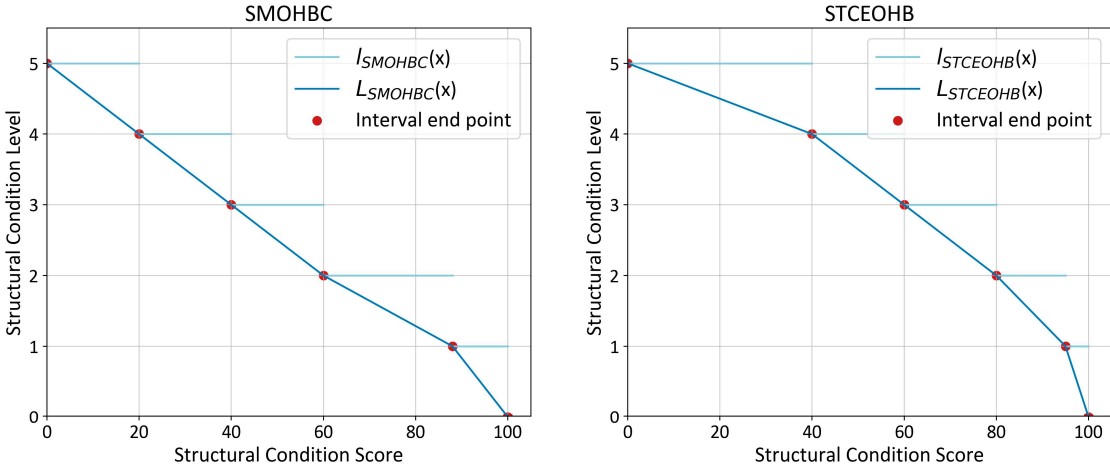

**Figure 3.** Continuous and Discrete Structural Condition Level Functions Under Different Scoring Systems.

The creation of $L(x)$ lays the foundation for score conversion between the two systems via interpolation. The conversion process involves calculating the continuous structural condition level of a score under the JTGT H21-2011 scoring system using $L_{SMOHBC}(x)$. Subsequently, the score under the JTG H11-2004 scoring system is determined based on the calculated structural condition level using $L_{STCEOHB}(x)$. For instance, if a score of 70 under the JTGT H21-2011 scoring system corresponds to a structural condition level of 1.643, then $L_{STCEOHB}(x)$ yields 85.357 as the converted value under the JTG H11-2004 scoring system. An example of this process is illustrated in Figure 4. In the figure, the red points still represent the interval end points, and the green points stand for the conversion example introduced above.

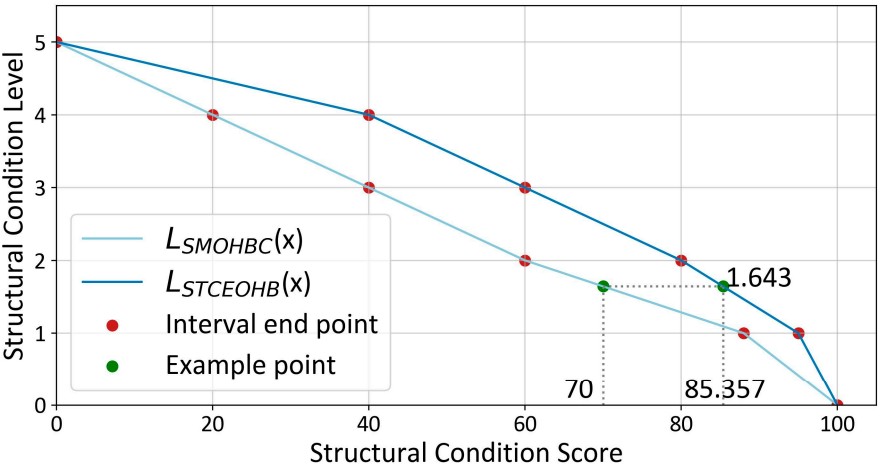

**Figure 4.** Example Demonstrating Score Conversion Between Scoring Systems.

The efficacy of these conversion rules is measured by the deviation before and after score conversion, quantified using the coefficient of deviation (*Err*) as defined in Equation (5).

$$Err = \frac{\left|level_{orig} - level_{conv}\right|}{5} \tag{5}$$

In Equation (5), $level_{orig}$ represents the structural condition level corresponding to the structural condition score of the bridge before conversion, and $level_{conv}$ represents the structural condition level corresponding to the structural condition score of the bridge after conversion. This coefficient evaluates the reasonability of the conversion rules, with the normalized absolute deviation of the structural condition level before and after conversion indicating their effectiveness. *Err* is employed to assess the rationality of the conversion rule in the subsequent chapter.

## 4. Case Study

In this study, a dataset comprising inspection reports and design drawings of 812 main-highway bridges from three highway routes in Shandong Province from 2010 to 2023 was collected, as shown in Table 2.

**Table 2.** Data Description.

| Group | Number of Bridges | Inspection Reports | Build Time |
|---|---|---|---|
| 1 | 220 | 2012, 2014, 2016, 2019 | 28 December 2008 |
| 2 | 343 | 2010, 2011, 2014, 2017, 2020 | 30 April 2007 |
| 3 | 249 | 2012, 2014, 2015, 2017, 2020 | 15 December 2007 |
| Total | 812 | 2010–2020 | 2007–2008 |

The data underwent extraction, cleaning, and integration processes, leading to the formation of a comprehensive regional bridge information database, as illustrated in Table 3.

**Table 3.** Examples of Bridge Attributes in the Regional Bridge Information Database.

| Feature | Meaning of the Feature | Example Value |
|---|---|---|
| Region Name | The region where the bridge is located | Beihuan |
| Bridge Name | Name of the bridge | K0+975.177 Tangwang Viaduct |
| Bridge Length | Length of the bridge | 8 m |
| Number of Spans | Span number of the bridge | 1 |
| Bridge Width | Width of the bridge | 11.75 m |
| Structure Type | The structure type of the bridge | Simply supported beam |
| Material | Material of the bridge | Reinforced concrete |
| Section type | The Cross-section type of the girder | Slab beam |
| Bearing Type | The type of the bearing | Plate rubber bearing |
| Pier Type | The type of the pier | Double column pier |
| Abutment Type | The type of the abutment | Light abutment |
| Footing Type | The type of the footing | Drilling piles |
| Pavement Type | The type of the pavement | Asphalt concrete |
| Expansion Joint Type | The type of the expansion joint | Maurer expansion joints |
| Build Time | The date when the bridge was built | 28 December 2008 |
| Designed code | The code that the bridge followed during the design process | General Code for Design of Highway Bridges and Culverts JTG D60-2004 |
| Design load | The form and value of the load that the bridge were designed to bear | Highway Level 1 |
| Design office | The company responsible for the design of the bridge | Shandong provincial communications planning and design institute group, Jinan, China |
| ADTT | Annual Average Daily Truck Traffic | 2022:23,150 |
| Deflection history | Deflection evolution of the bridge | Unknown |

Initially, inspection data were extracted and cleaned from the inspection report documents stored in WORD format. The scores and ratings of structural conditions for both bridges and their individual components were extracted, cleaned, and subsequently stored, as detailed in Table 4.

**Table 4.** Sample Extracted Structural Condition Scores for Bridges.

| Bridge Name | Super Structure Score | Substructure Score | Deck Score | Overall Score | Component Scores |
|---|---|---|---|---|---|
| K0+975.177 Tangwang Viaduct | 95.5 | 92.5 | 98.3 | 94.9 | . . . |

To address mismatched data errors present in inspection reports, the cleaning method described earlier was employed. Following cleaning, the coefficient of deviation was calculated, as depicted in Figure 5.

Figure 5 showcases the distribution of deviation coefficients defined in Equation (5) after the conversion of different bridges. The distribution reveals that most bridges exhibit small deviation coefficients, with nearly 95% having coefficients within 0.05. This distribution further substantiates the rationality of the cleaning conversion rule proposed earlier.

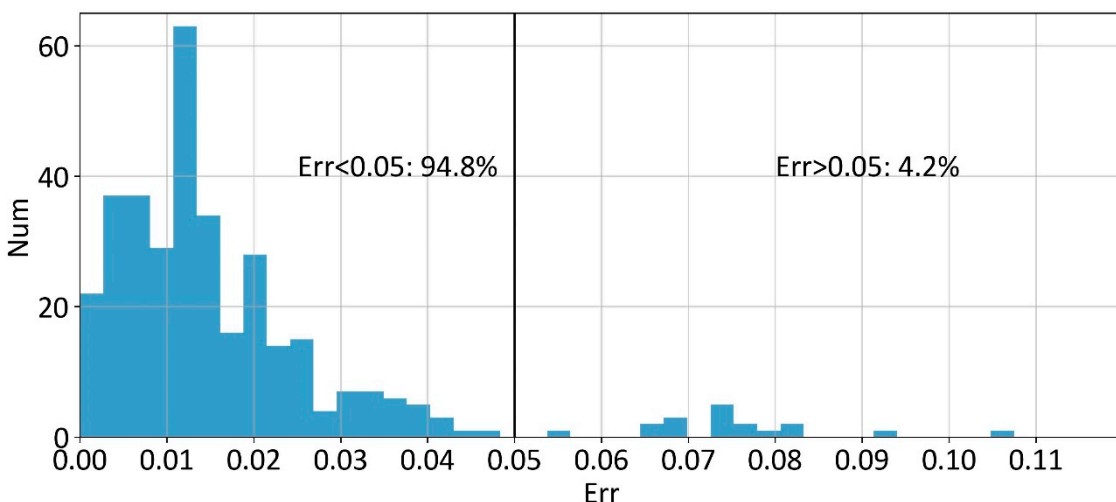

**Figure 5.** Distribution of Deviation Coefficients for Converted Scores After Data Cleaning.

This study also delves into exploring and analyzing various useful statistical features of attributes within the bridge network database. For instance, Figure 6 visually represents the distribution of span lengths among typical bridges in the region. The observation deduced from this distribution is that the majority of bridges with a span length of 40 m or less fall under the category of small and medium span bridges, which often share similar structural properties.

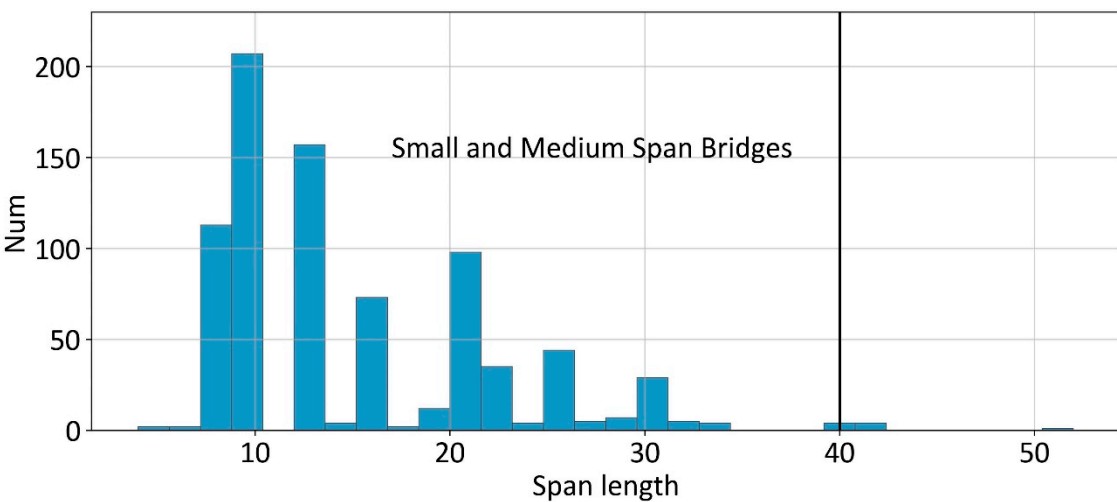

**Figure 6.** Distribution of Span Lengths for Bridges in the Dataset.

Table 5 offers insight into the distribution of bridge types within the regional bridge dataset. Beam bridges emerge as the dominant structure type among the region's bridges, particularly prevalent for small and medium span bridges. Furthermore, Figure 7 illustrates the prevalent section forms for beam bridges. Among the investigated bridges in Shandong Province, box girder and slab girder section forms are the most frequently used, aligning with the span distribution data depicted in Figure 6.

**Table 5.** Distribution of Bridge Types in the Dataset.

| Bridge Type | Arch Bridge | Culvert | Beam Bridge | Other |
| --- | --- | --- | --- | --- |
| Length | 0.048 km | 0.021 km | 62,045 km | 0.120 km |

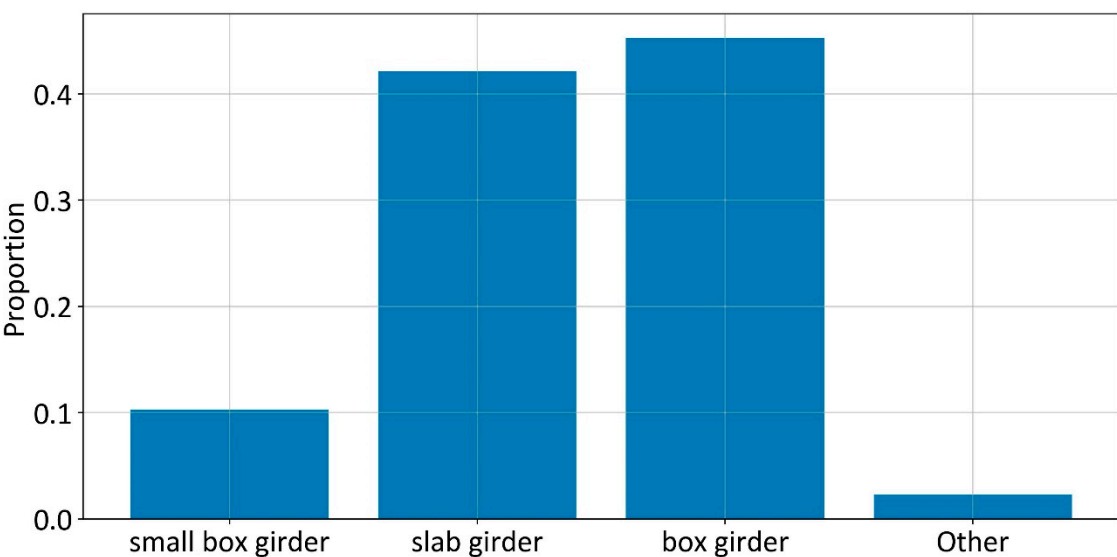

**Figure 7.** Distribution of Girder Section Types Among Beam Bridges in the Dataset.

Given the extensive historical structural health data that have been integrated, it holds significance to delve into the evolving patterns of structural conditions for regional bridges over time. Figure 8 serves to illustrate the distribution of structural condition scores for bridges across different years. As the age of bridges increases, the distribution of structural condition scores gradually shifts to the left, indicating an ongoing deterioration trend observed in regional bridges.

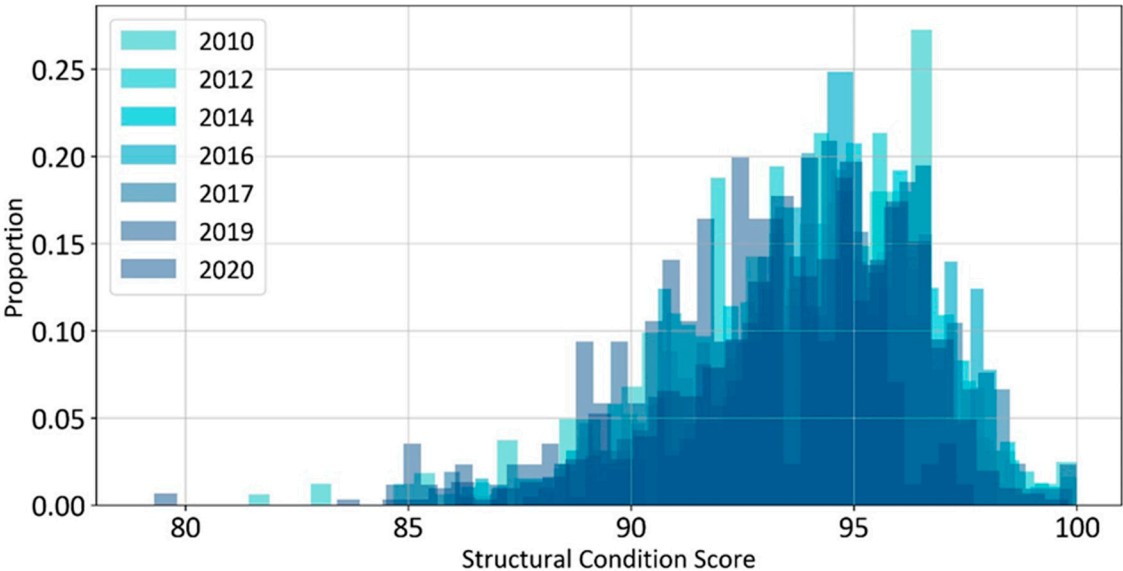

**Figure 8.** Changes in Distribution of Bridge Condition Scores Over Time.

Figure 9, on the other hand, offers insight into the distribution of structural condition levels for bridges within a specific region. This distribution further underscores the consistent deterioration trend, evident as the proportion of bridges classified under level 1 decreases with increasing bridge age. It is notable that due to regular maintenance efforts, bridges with structural condition levels below level 2 are quite rare—a trend observed across most bridges in Shandong Province.

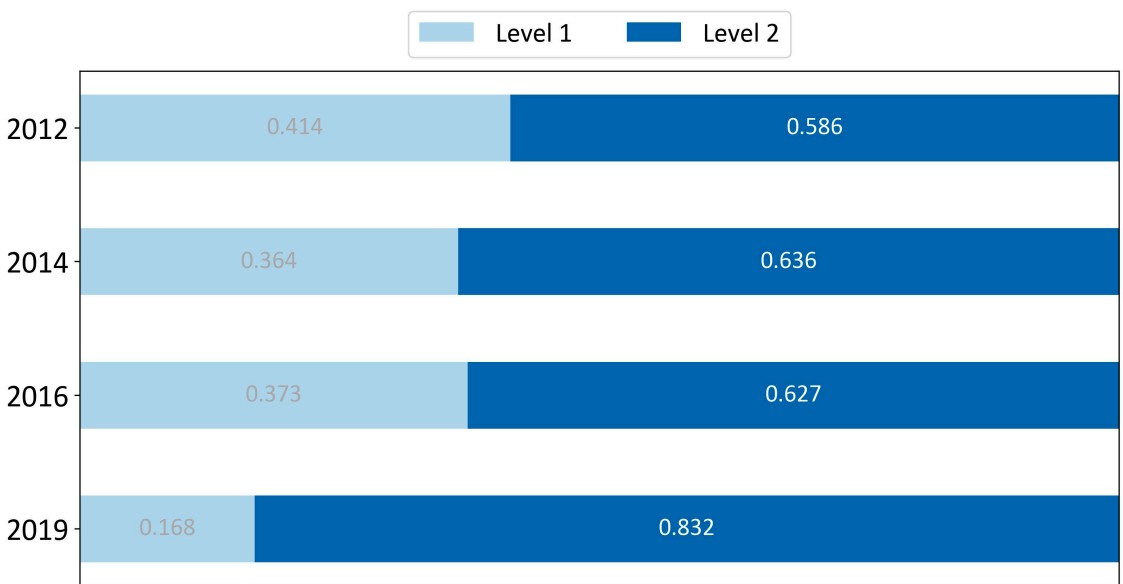

**Figure 9.** Changes in Distribution of Bridge Condition Levels Over Time.

## 5. Discussion

This study presents a pioneering approach to the extraction, cleaning, and integration of data, culminating in the creation of a comprehensive regional bridge database. A key innovation is the processing of unstructured inspection data from reports, a historically challenging area in structural condition assessment. Unlike previous research, which has mostly focused on road pavements [29,30], this study targets bridge inspection data, implementing systematic procedures to address prevalent errors such as "record errors" and "mismatched data." This enhances the reliability and precision of the inspection data.

Incorporating multi-source data, including inspection records, traffic statistics, and design and construction details, has proven effective in our case study. This integrated approach has the potential to transform bridge condition assessment, providing a solid base for informed decision making and strategic planning. The case study emphasizes the importance of meticulous data cleaning and integration for regional bridges, which is crucial for successful future data mining initiatives. This study addresses a research gap by detailing the form, acquisition methods, integration rules, and information data structure for multi-source regional bridge data.

This study can have worldwide benefits for the following reasons. First, although the conversion rules of rating scores were validated using data from Shandong province, its core approach—identifying intersections and invariants between different protocols—holds universal value. This principle is significant for an international audience, offering a systematic methodology adaptable to bridge assessment standards worldwide and aligning with evolving standards.

Beyond just score conversion, the overall vision of this paper encompasses establishing a comprehensive multi-source information database for regional bridges. The outlined methods for data extraction, cleaning, and integration are globally applicable. For instance, the selected attributes for the regional bridge database are relevant to most bridges worldwide. The identification of Record Errors in inspection reports and the proposed cleaning methods are universally applicable, providing a valuable toolkit for bridge assessment professionals globally.

In conclusion, while this study's immediate application is within the Chinese context, the strategic approach and data management techniques presented have broad relevance. They offer valuable insights and reference points for bridge condition assessment protocols in other regions of China and countries around the world.

However, this study is not without its limitations. The current extraction method primarily focuses on tabular data within inspection reports and does not account for

human-generated errors or data manipulation within these reports. Future research could enhance the robustness of the method by incorporating Natural Language Processing (NLP) technologies to interpret and comprehend inspection reports more accurately. Additionally, efforts could be directed toward assessing the authenticity of inspection data and exploring various avenues for utilizing the data stored in the established database, such as optimizing maintenance strategies for regional bridges.

## 6. Conclusions

In the context of assessing the condition of small and medium span bridges within a region, this paper has thoroughly examined methods for extracting, cleaning, and integrating multi-source information. It further presents the architecture of a regional bridge information database, which serves as a foundational data structure for subsequent research endeavors. The key conclusions drawn from this study are as follows:

1.  This study presents a groundbreaking method for automatically extracting data from unstructured bridge inspection reports, such as Word documents. This innovation is globally relevant, offering a way to efficiently use large amounts of previously untapped data, improving bridge analysis and infrastructure management worldwide.
2.  This study identified and rectified 'record errors' and 'mismatched data' in bridge inspection data, which has worldwide relevance. The developed cleaning methodologies ensure more accurate and harmonized datasets, applicable globally. Such advancements in data precision are crucial for effective bridge management and safety in various regions, transcending local codes and practices.
3.  This research introduces a comprehensive data integration approach that encompasses multiple information sources, including inspection data, traffic data, as well as design and construction drawings. By creating a unified bridge archive database, it overcomes the limitations of isolated data at the individual bridge level. This unified approach is vital for worldwide infrastructure management, as it facilitates advanced data mining and deterioration modeling applicable to bridges in various regions, enhancing global bridge safety and maintenance strategies.

**Author Contributions:** Conceptualization, G.X. and Y.X.; methodology, W.Z.; software, F.L.G.; validation, Y.X.; formal analysis, Y.X.; investigation, G.X.; resources, G.X.; data curation, G.X.; writing—original draft preparation, G.X.; writing—review and editing, G.Z.; visualization, G.X.; supervision, Y.X.; project administration, G.X., G.Z., and W.Z.; funding acquisition, G.X. All authors have read and agreed to the published version of the manuscript.

**Funding:** This paper was supported by the Transportation Science and Technology Program of Shandong Province (2021B51), and National Natural Science Foundation of China (52278313).

**Data Availability Statement:** Data are unavailable due to privacy restrictions.

**Conflicts of Interest:** The authors declare no conflict of interest.

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
