# Peer review of "A Practical Data Extraction, Cleaning, and Integration Method for Structural Condition Assessment of Highway Bridges"

_infrastructures, doi:10.3390/infrastructures8120183_

Round 1
Reviewer 1 Report
Comments and Suggestions for Authors
I cannot understand its English. The authors used difficult to understand vocabularies and I could not understand it well.
I will reject this paper.
Author Response
Dear Reviewer,
We sincerely appreciate the time and effort you invested in reviewing our manuscript. We wholeheartedly accept your critique regarding the language of the paper.
We understand the paramount importance of readability in conveying research findings effectively. To address this concern, we are committed to enhancing the language and expression in our paper to ensure it becomes more accessible and understandable to a broader readership. To achieve this, we plan to seek the assistance of professional native English speakers to refine the clarity and readability of our work.
Once again, we extend our gratitude for your valuable feedback and suggestions. Your insight is of great importance and will contribute to an improved quality of our paper. Thank you for your support.
Reviewer 2 Report
Comments and Suggestions for Authors
This paper proposed a practical framework for preprocessing multi-source raw data and establish a standard database aiming at regional highway bridges' condition assessment. This framework is very meaningful for this field. The whole study is rigid and well present. I believe it should be accepted and published in this journal.
Author Response
Dear Reviewer,
We sincerely appreciate your thoughtful evaluation of our work. Your recognition of the practical framework we proposed is immensely encouraging. We are gratified that you find the framework meaningful for this field.
Your positive assessment of our study being rigorous and well-presented is especially motivating. We are pleased to hear that you believe our paper merits acceptance and publication in this journal.
Your feedback and support are highly valued, and we look forward to contributing further to the field of regional highway bridge assessment.
Thank you for your time and consideration.
Reviewer 3 Report
Comments and Suggestions for Authors
The authors present an analytical contribution to the study of bridge managements systems. This reviewer has the following comments:
101. This reviewer considers that the evolution of the number of heavy-duty vehicles supported by the bridge should be a major concern in the data base. Light vehicles traffic is irrelevant.
202. Another data concern is the structural code and load specifications used for the design of the bridges.
303. This reviewer would also include the design office company that projected the structure, since design offices tend to replicate past designs.
404. It appears from Figure 7 that most bridges considered are concrete bridges: precast beam decks, slab girder decks and box girder decks.
505. It appears from the paper Figure 8 that most bridges have less than 20 years since construction. It would add to the paper to include a Figure similar to Fig. 8 with the histogram of the year of completion of the bridges.
606. This reviewer recommends in her lectures that bridges are topographically leveled in the inspections, so as to follow the historical evolution of foundation settlements and deck deflections. This appears to be basic to understand the evolution of the state of the bridge.
707. This reviewer misses in Figure 2 the deflection evolution of the bridge.
808. The very high coefficients in Fig. 8 denote that bridges are pretty young. Bridges are designed for a service life of 100 years. It would be more informative to have bridges in the database of 50-75-100 years of age.
909. Overall, the topic of bridge management systems has much present and future, and I think the paper is worth for publication in MDPI Infrastructures.
110. Please correct the 94.6 duplication in lines 311 and 312.
Round 2
Reviewer 1 Report
Comments and Suggestions for Authors
Please see the attachment
